# Pharmacokinetics and Pharmacodynamics of Colistin Methanesulfonate in Healthy Chinese Subjects after Multi-Dose Regimen

**DOI:** 10.3390/antibiotics11060798

**Published:** 2022-06-14

**Authors:** Yaxin Fan, Yi Li, Yuancheng Chen, Jicheng Yu, Xiaofen Liu, Wanzhen Li, Beining Guo, Xin Li, Jingjing Wang, Hailan Wu, Yu Wang, Jiali Hu, Yan Guo, Fupin Hu, Xiaoyong Xu, Guoying Cao, Jufang Wu, Yingyuan Zhang, Jing Zhang, Xiaojie Wu

**Affiliations:** 1Institute of Antibiotics, Huashan Hospital, Fudan University, Shanghai 200040, China; fanyaxin@fudan.edu.cn (Y.F.); 12220220011@fudan.edu.cn (Y.L.); liuxiaofen@huashan.org.cn (X.L.); 19111220031@fudan.edu.cn (W.L.); guobeining@huashan.org.cn (B.G.); 18111220016@fudan.edu.cn (X.L.); wuhailan@huashan.org.cn (H.W.); 20111220007@fudan.edu.cn (Y.W.); hujiali@huashan.org.cn (J.H.); guoyan@fudan.edu.cn (Y.G.); hufupin@fudan.edu.cn (F.H.); 20111240001@fudan.edu.cn (X.X.); zhangyingyuan@huashan.org.cn (Y.Z.); 2Key Laboratory of Clinical Pharmacology of Antibiotics, National Population and Family Planning Commission, Shanghai 200040, China; chenyuancheng@huashan.org.cn (Y.C.); yujicheng@fudan.edu.cn (J.Y.); wangjingjing@huashan.org.cn (J.W.); caoguoying@huashan.org.cn (G.C.); wujufang@huashan.org.cn (J.W.); 3National Clinical Research Center for Aging and Medicine, Huashan Hospital, Fudan University, Shanghai 200040, China; 4Phase I Clinical Research Center, Huashan Hospital, Fudan University, Shanghai 200040, China

**Keywords:** colistin methanesulfonate, colistin, pharmacokinetics, pharmacokinetics/pharmacodynamics, human subjects

## Abstract

Colistin methanesulfonate (CMS) is an important treatment option for infections caused by carbapenem-resistant Gram-negative organisms (CROs). This study evaluated the pharmacokinetic/pharmacodynamic (PK/PD) profiles and safety of CMS in Chinese subjects following a recommended dosage. A total of 12 healthy Chinese subjects received CMS injections at 2.5 mg/kg once every 12 h for 7 consecutive days. The PK/PD profiles of the active form of CMS, colistin, against CROs were analyzed with the Monte Carlo simulation method. No serious adverse events were observed. The average steady-state plasma concentrations of CMS and colistin were 4.41 ± 0.75 μg/mL and 1.27 ± 0.27 μg/mL, and the steady-state exposures (AUC_0–12,ss_) were 52.93 ± 9.05 h·μg/mL and 15.28 ± 3.29 h·μg/mL, respectively. Colistin, at its minimum inhibitory concentration (MIC) of 0.5 μg/mL, has >90% probability to reduce CROs by ≥1 log. The PK/PD breakpoints for the ≥1 log kill were ≥MIC_90_ for carbapenem-resistant *Klebsiella pneumoniae* and *Pseudomonas aeruginosa*, but were ≤MIC_50_ for carbapenem-resistant *Acinetobacter baumannii*. The recommended dose regimen of CMS for 7 consecutive days was safe in Chinese subjects. The systemic exposure of colistin showed a high probability of being sufficient for most CROs, but was not sufficient for some carbapenem-resistant *A. baumannii*.

## 1. Introduction

Carbapenem-resistant Gram-negative organisms (CROs), including carbapenem-resistant *Klebsiella pneumoniae*, *Acinetobacter baumannii* and *Pseudomonas aeruginosa*, are important pathogens that are threatening public health [1]. In recent years, infections caused by CROs have been increasing dramatically all over the world, including China. In 2020, the drug resistance surveillance and data collection conducted by China antimicrobial surveillance network showed that the detection rate of CROs has increased two to eight times over the years since 2005 [2]. The detection rates of imipenem-resistant *K. pneumoniae*, *A. baumannii* and *P. aeruginosa* were 23.2%, 68.1% and 23.2%, respectively [2]. Increased morbidity, mortality and longer hospital stays were observed in patients infected with CROs compared with that caused by carbapenem-sensitive bacteria [3,4]. The drugs available to treat CRO infections are extremely limited, and polymyxins are considered as the “last line of defense” for the treatment of CRO infections [5]. Colistin methanesulfonate (CMS) is one of the polymyxins, and it is a pro-drug that converts into the active form of colistin after entering the human body.

The ratio of the area under the free colistin plasma concentration–time curve from 0 to 24 h (*f*AUC_0–24_)/minimum inhibitory concentration (MIC) is considered as the best pharmacokinetic/pharmacodynamic (PK/PD) index for the efficacy of CMS [6]. A study performed in animal infection models showed that when *f*AUC_0–24_/MIC was at about 12, which was equivalent to an average steady-state plasma concentration (C_avg,ss_) of colistin at 1 μg/mL, resulted in at least one log kill of *A. baumannii* and *P. aeruginosa* [6]. CMS in clinic has generally been expressed as the internationally accepted term of colistin base activity (CBA) or the international unit (IU) [7]. For patients with normal renal function (creatinine clearance ≥ 80 mL/min), the European Medicines Agency (EMA) recommends 9MIU CMS (approximately 300 mg CBA), and the US Food and Drug Administration (FDA) recommends a dose of 2.5–5 mg/kg CBA [8]. Studies have demonstrated that at the recommended doses of EMA and FDA, only 65–75% of patients with normal renal function achieved C_avg,ss_ at ≥1 μg/mL [8]. The most important safety issue is nephrotoxicity during the clinical application of CMS. The colistin plasma concentration is an indicator of nephrotoxicity; thus, it is recommended to monitor the colistin plasma concentration during CMS treatment for nephrotoxicity. In 2019, the International Consensus Guidelines for the Optimal Use of the Polymyxins recommended that to achieve the maximum therapeutic effect, the steady-state exposure, the area under the concentration–time curve from 0 to 24 h (AUC_0–24,ss_) of the active ingredient colistin at 50 h·μg/mL, C_avg,ss_ needs to be maintained at 2 μg/mL. Exceeding this exposure would increase the incidence and severity of acute nephrotoxicity [7]. Therefore, obtaining the steady-state plasma concentration of colistin following the recommended multi-dose administration of CMS is important for evaluating the efficacy and safety of CMS in the treatment of Chinese patients with CRO infections.

A new CMS formula was produced with a 4:1 ratio of CMS A:CMS B, and met the requirements of the European Pharmacopoeia EP9.2 [9]. Preliminary studies revealed that different component contents in CMS preparations had a big impact on the exposure of colistin [10]. The C_avg,ss_ (95% CI) was 0.92 (0.85, 0.99) μg/mL after 7 days of 2.5 mg/kg CBA every 12 h (q12 h) administration of this new formulation [10], as simulated by the obtained single-dose PK parameters [10]. The purpose of this study was to investigate the PK characteristics of CMS and colistin in plasma and urine following multiple intravenous administrations of the new CMS formula at the recommended multi-dose regimen (2.5 mg/kg CBA q12 h) in healthy Chinese subjects. A further analysis of the PK/PD of colistin against CROs obtained from Chinese patients was performed to evaluate the efficacy of this dosing regimen. The results in healthy subjects provided essential pharmacological and safety information for the product and a basis for dosing regimens for patients with CRO infections.

## 2. Results

### 2.1. Demographics of the Subjects

Twelve subjects, six males and six females, enrolled in this study. All subjects had completed the entire study according to the requirements of the protocol and nobody withdrew from the trial. Therefore, the data obtained from all the 12 subjects entered the pharmacokinetic and safety analysis. The average age of the 12 subjects was 23.50 ± 1.17 years, the average body weight was 61.79 ± 6.89 kg, the average body mass index (BMI) was 23.28 ± 2.03 kg/m^2^, the average serum creatinine was 63 ± 14 μmol/L and the average creatinine clearance rate (ClCr) was 131.54 ± 14.76 mL/min.

### 2.2. Pharmacokinetics Analysis

Figure 1 shows the average plasma concentrations of CMS and colistin after multiple intravenous infusions of CMS at 2.5 mg/kg q12 h for 7 consecutive days (day 1 and day 7: one dose; day 2 to day 6: q12 h). The calculated plasma PK parameters of CMS and colistin are shown in Table 1. After 7 days of CMS infusions, the C_avg,ss_ for CMS and colistin were 4.41 ± 0.75 μg/mL and 1.27 ± 0.27 μg/mL, respectively; the AUC_0–12,ss_ for CMS and colistin were 52.93 ± 9.05 h·μg/mL and 15.28 ± 3.29 h·μg/mL, respectively. The differences between the first and the last doses in all the PK parameters of colistin were statistically significant (*p* < 0.05). The AI after 7 consecutive days of multiple dosing was 1.07 ± 0.10 for CMS, indicating no obvious accumulation of the pro-drug CMS in human bodies; the AI was 1.32 ± 0.06 for the active formed colistin, indicating that a slight accumulation of colistin occurred in the human bodies.

Figure 2 shows the average urine excretion rates of CMS and colistin in 12 subjects after multiple doses of 2.5 mg/kg CBA were administrated intravenously. The cumulative urine excretion rates of CMS within 0–24 h were 72.54 ± 12.28% and 54.97 ± 14.64% after the first and last doses, respectively. The cumulative urine excretion rates of colistin within 0–24 h were 0.41 ± 0.28% and 1.22 ± 0.70% after the first and last doses, respectively.

### 2.3. Pharmacokinetics/Pharmacodynamics Analysis

Based on the published PK/PD index as the target, PTA values for each colistin MIC to reduce one log or two log bacterial counts were determined using the Monte Carlo simulation method (Table 2) for the three types of bacteria. The results showed that to achieve a PTA of >90%, colistin MIC was at ≤0.5 μg/mL for *K. pneumoniae* and *A. baumannii*; and colistin MIC was at ≤1 μg/mL or 0.5 μg/mL for *P. aeruginosa* in reducing one log or two log bacterial counts, respectively.

Next, the PK/PD breakpoints were compared with their corresponding published colistin MIC_50_ and MIC_90_ in the three types of CROs (Table 3). The results showed that the PK/PD breakpoint of 0.5 µg/mL in killing one-log CRKP or CRPA was greater or equal to their MIC_90_, indicating that most CRKP and CRPA were likely to be sensitive to this recommended CMS administration regimen. In contrast, the colistin PK/PD breakpoint of 0.5 µg/mL in killing one- or two-log CRAB was no more than MIC_50_, indicating that some CRAB may have not been sensitive to this recommended CMS administration regimen.

### 2.4. Safety Evaluation

No deaths or other serious adverse events occurred during the study. None of the major adverse events were severe enough to require an early termination of the trial. No abnormal vital signs nor QTc prolongations or other clinically significant electrocardiogram abnormalities were found. Eleven out of the twelve subjects showed various adverse effects after the multi-dose intravenous infusions of CMS. Detailed symptoms, incidences and their durations in individual subjects are presented in Table 4. All the clinical adverse events were mild and transient. During the drug administration period, there was no increase in urine β2 microglobulin and urine NGAL, and no nephrotoxicity was observed, indicating that the safety of this recommended multi-dose administration of CMS was acceptable in the Chinese subjects. Changes in early renal damage indicators, serum creatinine and urea are shown in Appendix A.

## 3. Discussion

This study evaluated the applicability of CMS at the recommended dosing regimen (2.5 mg/kg CBA q12 h) for the first time in Chinese subjects. Our results showed that after multiple intravenous administrations at the recommended CMS dose, the C_avg,ss_ of colistin in plasma was 1.27 ± 0.27 μg/mL, which was higher than of the efficacious level of 1 μg/mL [8], and expected to kill one-log *A. baumannii* and *P. aeruginosa* in animal infection models [6]. Nation et al. [8] found that the plasma C_avg,ss_ of colistin in 51 critically ill patients with normal renal function (CLCr ≥ 80 mL/min) was 1.29 (0.45–5.28) μg/mL based on the FDA-recommended dose (2.5–5 mg CBA/kg), and the median concentration in 214 patients with different renal functions was 2.35 (0.24–9.92) μg/mL [13]. Kristoffersson et al. [14] included 87 critically ill patients with CrCL >120 mL/min, and after the administration of a loading dose of 9 MIU, followed by a 4.5 MIU twice-daily maintenance dose, patients had a median C_avg,ss_ concentration of 1.6 (0.4–4.8) μg/mL. The observed C_avg,ss_ in healthy subjects was quite similar to those that were administrated according to the CMS algorithm in critical ill patients, suggesting that the current dosage could be applicable to the critical ill patients in China. The PK data obtained from this study suggested that the recommended dose of CMS was sufficient to achieve the efficacious colistin concentration in human plasma against most CROs.

This study showed that the PK/PD breakpoint was greater than or equal to its MIC_90_ when killing one log of CRKP or CRPA, suggesting that it may be sensitive to the current dosing regimen. Jitaree et al. [15] performed a PK/PD analysis of CRKP and *Escherichia coli* based on PK data in 116 critically ill patients. For patients with creatinine clearance ≥80 mL/min, a 150 mg CBA q12 h regimen supported CRKP infection with an MIC of 0.5 μg/mL, but the cumulative fraction of response was only 12.45%, which may be due to the inclusion of only 22 CRKP strains and the high colistin MIC value (ranging from 4 to >128 μg/mL). Sorlí et al. [16] calculated an AUC_0–24_/MIC of 60.5 ± 56.4 and an *f*AUC_0–24_/MIC of 30.2 ± 28.2 in 33 patients with extremely drug-resistant (XDR) *P. aeruginosa* urinary tract infections (UTIs), with only 29.4% of patients achieving the optimal PK/PD target (AUC_0–24_/MIC ≥ 60). Patients had a C_avg,ss_ of 1.19 (0.21–5.20) μg/mL and 58.8% showed plasma concentrations of colistin higher than the MIC of the isolated *P**. aeruginosa*. The low PK/PD attainment may be associated with high target values (C_avg,ss_ = 2.5 μg/mL, AUC_0–24_/MIC = 60), which may exceed the nephrotoxicity threshold of the 2019 international consensus guidelines [7].

The calculated PK/PD breakpoint for the one-log or two-log killing of *A. baumannii* was 0.5 μg/mL, which was ≤MIC_50_ of colistin for CRAB [12]. The result suggested that when using CMS to treat CRAB-induced infections as a single-agent treatment, the anti-bacterial effects could only be expected in CRAB with an MIC_90_ of colistin at ≤0.5 μg/mL. For the patients infected by CRAB with MIC_90_ > 0.5 μg/mL, combination treatments of CMS with other synergistic anti-bacterial drugs should be considered [17,18,19,20]. The International Consensus Guidelines recommend the colistin combination treatment for invasive infections caused by CROs [7]. A previous study [17] showed that colistin combined with meropenem, minocycline or rifampicin had a synergistic effect on XDR *A. baumannii* isolates obtained from Chinese patients in vitro. Although a synergistic effect was observed in the combination of meropenem and colistin in CRAB isolates obtained from Chinese patients using a PK/PD model [18], this combination did not show higher rates of clinical success in severe *A. baumannii* infections [21]. In addition, studies showed that colistin plus sulbactam was synergistic against CRAB using a PK/PD model [19]. In a network meta-analysis of 29 studies with 2529 patients with infections caused by multi-drug-resistant and XDR *A. baumannii*, the combination of colistin, sulbactam and tigecycline had the highest clinical cure rate [22]. Compared with the combination of tigecycline and colistin monotherapy, colistin combined with sulbactam was associated with a higher microbiological cure rate [22]. Thus, it was expected that the current recommended dose (CMS at 2.5 mg/kg CBA q12 h) combined with other antibiotics, such as sulbactam, may achieve effective microbiological efficacy in Chinese patients infected with CROs.

The PK parameters of CMS and colistin from the current study were compared to those from a paper published in 2018, where a different CMS formula was used [23]. The two formulas had the same total amount of the main components of CMS, approximately 80%, but the ratio of CMS A:CMS B was updated from 1:2 [23] to 4:1 (current study). With the updated formula, the exposure of colistin increased considerably, with the AUC_0–12,ss_ changing from 9.23 ± 1.79 h·μg/mL [23] to 15.28 ± 3.29 h·μg/mL, and the C_avg,ss_ changing from 0.760 ± 0.151 μg/mL [23] to 1.27 ± 0.27 μg/mL. Study [24] showed that healthy Japanese subjects receiving intravenous infusions of CMS at 2.5 mg/kg CBA twice a day for 2.5 days resulted in AUC_0–12,ss_ of 25.70 ± 7.49 h·μg/mL, which was higher than what was obtained in the healthy Chinese subjects in this study. A possible reason for the difference might be the formulation difference in the Japanese study.

In this study, the steady-state urine excretion rates of CMS during the 0 to 24 h post-administration period in the healthy Chinese subjects after the last dose were 54.97 ± 14.64%. The average steady-state urine excretion rates of CMS at 42.5% and 33.1% were obtained in patients with multi-drug-resistant Gram-negative bacterial infections, who had a glomerular filtration rate (eGFR) of ≥80 mL/min and <80 mL/min during the 0 to 6 h post-administration period [25], and the results of the patients with normal renal functions were close to the results obtained in this study. Due to CMS being converted into the active form of colistin in the kidneys and bladder, the concentrations of colistin measured within the first 9 h post-administration from the 12 subjects in this study ranged between 1.14 and 124 μg/mL; thus, CMS can be used to treat UTIs caused by CROs. A clinical study was conducted [16] including 33 patients with urinary tract infections (24 cases of lower UTI and 9 cases of pyelonephritis) caused by XDR *P. aeruginosa*. The main treatment option was CMS monotherapy, and some were CMS combined with meropenem, amikacin or ceftazidime given at a daily dose of 2.21 ± 1.25 mg/kg CBA. After the CMS treatment, the bacteria were eradicated in 76.9% of patients.

Nephrotoxicity is a major safety concern in the clinical application of CMS, with colistin-induced acute kidney injury apparent within the first 72 h of treatment [26]. Studies have reported that the incidence of nephrotoxicity in CMS-treated patients was 26.0–74.1% [16]. In this study, healthy Chinese subjects received the recommended dose of CMS with no increase in urinary β2 microglobulin and NGAL after 7 days of dosing. No obvious nephrotoxicity occurred during the study; thus, the recommended CMS dose regimen is considered safe in healthy Chinese subjects. Yendewa et al. [27] also studied an increase in CMS dose from 2.5 mg/kg q12 h to q8 h without an increase in nephrotoxicity. In addition, colistin loading doses can rapidly reach target concentrations, but Katip et al. [28] found that loading doses had a higher risk of nephrotoxicity than non-loading-dose patients (adjusted HR, 1.57, 95% CI 1.14–2.17, *p* = 0.006). In a meta-analysis of eight clinical studies including 1115 subjects, no correlation between the loading dose and increased nephrotoxicity occurred (RR = 1.31; 95% CI = 0.90–1.91) [29]. Nevertheless, further study in the occurrence of nephrotoxicity in patients with CRO infections is needed when treated with the recommended CMS dose regimen.

This study had some limitations. First of all, the number of cases included in this study was small, and the PK/PD analysis was based on PK parameters obtained from healthy subjects. The PK parameters of patients may differ from that of healthy subjects, pending the future inclusion of patients with CRO infections in the PK/PD analysis to evaluate the attainment of this dosing regimen. Second, the urine samples collected on the first day of the test drug administration were stored at −70 ± 10 °C without pre-mix with an equal volume of 1% BSA. Therefore, the results from the urine samples collected on the first day were for reference only.

## 4. Materials and Methods

### 4.1. Study Design and Dosing Regimen

A single-center, open-label, multi-dose trial design was conducted following multiple intravenous administrations of CMS in 12 healthy Chinese subjects. The study was conducted in Huashan Hospital from February to May 2019. The study protocol was reviewed and approved by the Ethics Committee of Huashan Hospital, Fudan University (Shanghai, China), under number 2018–432.

Colistin methanesulfonate for injection was produced and provided by CHIA TAI TIAN-QING Pharmaceutical Group Co., Ltd. (Nanjing, Jiangsu, China). The potency of each bottle of the drug was 150 mg (calculated as the activity unit of CBA), which was equivalent to 312 mg CMS. The dose regimen was 2.5 mg/kg CBA for 7 consecutive days with 12 h between each administration, except for one dose on day 1 (first dose) and day 7 (last dose). Each intravenous infusion time was 90 ± 10 min.

### 4.2. Subjects

Healthy Chinese male and female subjects, who were 19 to 45 years old, with a body mass index of 19.0 to 26.0 kg/m^2^ were recruited into this study. Informed consent forms were voluntarily signed by all the subjects before being enrolled in the study. The body weights were ≥45 kg and ≥50 kg for female and male subjects, respectively. The female subjects had negative serum pregnancy test results and were not breastfeeding during the study. All subjects did not use drugs known to cause damage to certain organs or participated in other drug trials within 3 months prior to entering the study; all subjects had not used any drugs 2 weeks prior to the starting day of the study. All the subjects were asked for their medical history, evaluated with physical examination, vital signs, 12-lead electrocardiogram and clinical laboratory tests.

### 4.3. Sample Collection and Preparation for Pharmacokinetics Analysis

The blood samples were collected on day 1 and day 7 at pre-dose during intravenous infusion (30 min and 60 min after the start of intravenous infusion), immediately after the end of intravenous infusion (1.5 h) and at 1.75 h, 2 h, 2.5 h, 3.5 h, 5 h, 7 h, 9 h, 12 h, 16 h and 24 h after the start of intravenous infusion. From day 2 to day 6, blood samples were collected before and immediately after the end of infusion of the first dose. All blood samples were collected and transferred into an EDTA-K2 tube, and were then centrifuged at 3500 rpm for 10 min at 4 °C. Within 1 h after blood collection, the plasma samples were stored at −70 ± 10 °C until analysis [10].

Urine samples were collected on day 1 and day 7 at pre-dose, 0–3 h, 3–6 h, 6–9 h, 9–12 h, 12–16 h and 16–24 h after the start of intravenous infusion. Urine samples (700 μL) were mixed with equal volume of 1% bovine serum albumin (BSA) prior and stored at −70 ± 10 °C until analysis.

### 4.4. Determination of CMS and Colistin Concentrations in Plasma and Urine Samples

The concentrations of CMS and colistin in plasma and urine samples were analyzed with a verified ultra-performance liquid chromatography tandem mass spectrometry method [30]. To determine the free colistin concentration, the matrix was alkalized with 5% ammonia at 0–4 °C, and CMS concentrations were obtained with acid hydrolysis, and then loaded onto the Oasis weak cation exchange 96-well plate (WCX, Waters, Milford, MA, USA). The standard curves of colistin A/colistin B for the plasma and urine samples ranged from 0.0446/0.0332–4.46/3.32 μg/mL to 0.0223/0.0166–2.23/1.66 μg/mL [10], respectively. The concentrations of colistin and CMS were determined as the major components of colistin A+ colistin B and colistin A methanesulfonate (CMS A) + colistin B methanesulfonate (CMS B), respectively.

### 4.5. Pharmacokinetics Analysis

The PK parameters of CMS and colistin in plasma were calculated by the non-compartment model [23] using the WinNonlin8.0 software (Pharsight, Mountain View, CA, USA). The plasma PK parameters calculated after the first dose included peak concentration (C_max_), time to reach peak concentration (T_max_), half-life (T_1/2_), area under the plasma concentration–time curve from time 0 to 12 h (AUC_0–12_), AUC from 0 to 24 h (AUC_0–24_), total clearance (CL_t_) and distribution volume (V_d_). Following the last dose on day 7, the plasma PK parameters calculated included peak concentration at steady state (C_max,ss_), trough concentration at steady state (C_min,ss_), C_avg,ss_, T_max_, T_1/2_, AUC from time 0 to 12 h at steady-state (AUC_0–12,ss_), clearance at steady state (CL_ss_), distribution volume at steady state (V_ss_), accumulation index (AI) and the fraction of CMS converted to colistin in the body (*f*_m_). The urine excretion rate was calculated according to the correction of renal clearance (CL_r_). The PK parameters in urine samples calculated included interval urinary excretion rate, cumulative urinary excretion rate and CLr.

The PK parameters were expressed as average ± standard deviation (mean ± SD). The main PK parameters after the first and last doses were evaluated by paired t test (SAS 9.4, SAS Institute, Inc., Cary, NC, USA). *p* value ≤ 0.05 was considered as significant.

### 4.6. Pharmacokinetics/Pharmacodynamics Analysis

The target values of *f*AUC_0–24_/MIC used in this study were from published PK/PD study reports of colistin, conducted on animal infection models. The target value of *f*AUC_0–24_/MIC was ≥17.4 for 1-log kill of *K. pneumoniae* [11]; the target value of *f*AUC_0–24_/MIC was ≥13.9 or 17.6 for 1-log or 2-log kill of *A. baumannii*, respectively [6]; the target value of *f*AUC_0–24_/MIC was ≥10.9 or 13.7 for 1-log or 2-log kill of *P. aeruginosa*, respectively [6]. The free colistin fraction *f* used was 50% [6,31]. AUC_0–24,ss_ values from this study were calculated as 2 times AUC_0–12_ on day 7 based on the twice-daily administration regimen.

The probability of target attainment (PTA) was determined using Monte Carlo simulation method based on the publish PK/PD index for each type of bacteria and AUC_0–24, ss_/MIC from this study. Monte Carlo simulations were performed using Excel macros (written in VBA) and MATLAB software (MathWorks, version 7.0.1, Natick, MA, USA). The macro would call the MATLAB program to perform the Monte Carlo simulation and output the calculation results to an Excel worksheet. A random number of 5000 AUC_0–24,ss_ was generated according to the optimal distribution, and then the colistin *f*AUC_0–24_/MIC was calculated and the percentage of *f*AUC_0–24_/MIC above the target value was counted. The highest colistin MIC values that achieved the PTA ≥ 90% were used as the PK/PD breakpoints.

The pharmacodynamic data of MIC used were from a published paper [12]. In this paper, 377 strains of carbapenem-resistant *K. pneumoniae* (CRKP), 317 strains of carbapenem-resistant *A. baumannii* (CRAB) and 346 strains of carbapenem-resistant *P. aeruginosa* (CRPA) were collected from Chinese patients, and MIC values of colistin to the above-mentioned bacteria were determined with the broth microdilution method. MIC_50_ and MIC_90_ were defined as the minimal colistin concentration to inhibit 50% and 90% of bacteria in growth, respectively.

The PK/PD breakpoint was used to evaluate the sensitivity of bacteria to colistin: the bacteria were considered as sensitive to colistin when the PK/PD breakpoint ≥ MIC_90_, and the bacteria were considered as not sensitive to colistin when the PK/PD breakpoint ≤ MIC_50_.

### 4.7. Safety Evaluation

During the study, close observations and detailed records were kept in all subjects for clinical adverse events (AE), abnormal results of clinical laboratory tests and other specific examinations. The early renal damage indicators, including serum β2-microglobulin, serum Cysteine dehydrogenase inhibitor C (Cystatin C), urine neutrophils gelatinase-associated lipocalin (NGAL) and urine β_2_-microglobulin, were tested at day 4 and day 8 after the start of drug administration.

## 5. Conclusions

In conclusion, the recommended multiple intravenous doses of CMS at 2.5 mg/kg CBA once every 12 h for 7 consecutive days was safe in healthy Chinese subjects. After multiple administrations, a slight accumulation of the active form of colistin was observed with the C_avg,ss_ at 1.27 ± 0.27 μg/mL. The PK/PD breakpoint of 0.5 µg/mL and 1 µg/mL to kill one-log CRKP or CRPA was greater than or equal to its MIC_90_; however, the PK/PD breakpoint of colistin to CRAB was ≤MIC_50_ (0.5 μg/mL), suggesting that the recommended dose of CMS may not be sufficient in treating some patients with high colistin MIC.

## Figures and Tables

**Figure 1 antibiotics-11-00798-f001:**
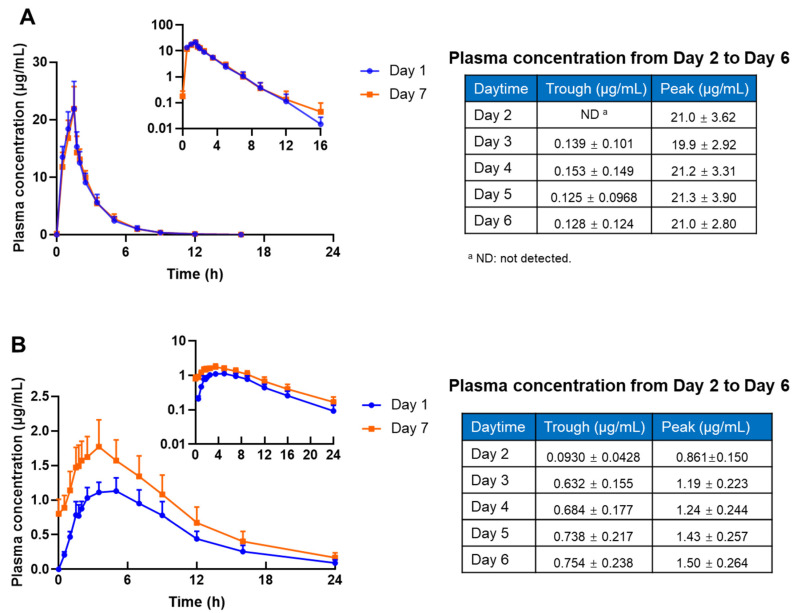
Mean plasma concentration–time profiles of CMS (**A**) and colistin (**B**) in healthy subjects after treatment with CMS injections at 2.5 mg/kg CBA 7 consecutive days (subjects were dosed every 12 h from days 2 to 6, with a single dose on days 1 and 7; data are expressed as the mean and standard deviation of 12 subjects).

**Figure 2 antibiotics-11-00798-f002:**
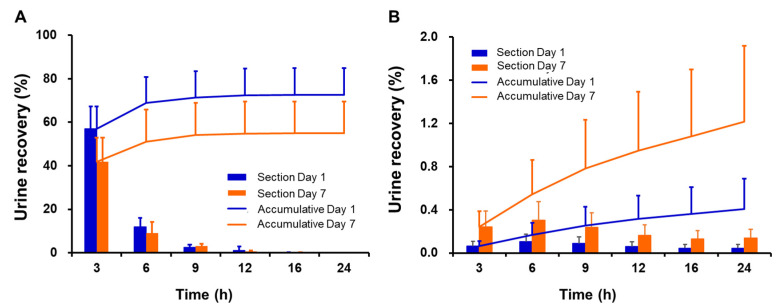
Mean urine excretion rate–time curves of CMS (**A**) and colistin (**B**) in healthy subjects after multiple doses of 2.5 mg/kg CBA CMS intravenous infusions. (The bar graph shows the interval segmented urine excretion rate, and the line indicates the cumulative urine excretion rate. The data are presented as the means and standard deviations from 12 subjects.)

**Table 1 antibiotics-11-00798-t001:** Summary of pharmacokinetic parameters in healthy Chinese subjects after multi-dose intravenous infusions of CMS (mean ± SD, n = 12).

PK Parameter	Unit	CMS			Colistin		
		Day 1 (First Dose)	Day 7 (Last Dose)	*p* Value *	Day 1 (First Dose)	Day 7 (Last Dose)	*p* Value *
C_max_	μg/mL	22.2 ± 4.38	21.9 ± 3.85	0.882	1.17 ± 0.18	1.79 ± 0.37	**<0.001**
C_min,ss_	μg/mL	\	0.133 ± 0.141	\	\	0.67 ± 0.22	\
C_avg,ss_	μg/mL	\	4.41 ± 0.754	\	\	1.27 ± 0.27	\
AUC_0–12_	h·μg/mL	53.73 ± 9.77	52.93 ± 9.05	0.636	9.80 ± 1.71	15.28 ± 3.29	**<0.001**
AUC_0–24_	h·μg/mL	53.96 ± 9.93	\	\	12.59 ± 2.55	\	\
T_max_ ^a^	h	1.35 ± 0.18	1.41 ± 0.13	0.414	4.42 ± 0.90	3.38 ± 0.68	**0.015**
T_1/2_ ^b^	h	1.37 ± 0.34	2.32 ± 1.68	0.083	5.15 ± 0.95	5.90 ± 0.58	**0.011**
CL_t_	L/(h·kg)	0.08 ± 0.02	0.08 ± 0.02	0.777	0.21 ± 0.05	0.18 ± 0.04	**<0.001**
CL_r_	L/(h·kg)	0.06 ± 0.01	0.04 ± 0.02	**0.008**	0.0008 ± 0.0004	0.0015 ± 0.0007	**<0.001**
V_d_	L/kg	0.15 ± 0.03	0.14 ± 0.01	0.367	1.52 ± 0.22	1.66 ± 0.25	**0.019**
AI	\	\	1.07 ± 0.10	\	\	1.32 ± 0.06	\
*f* _m_	\	0.27 ± 0.12	0.45 ± 0.15	**0.007**	\	\	\

Note: CMS doses at 2.5 mg/kg (CBA) once every 12 h for 7 consecutive days with the dose interval between the first and the second dose being 24 h. The PK parameters on day 7 were calculated as the steady state. Abbreviations: C_max_: peak concentration; C_min_: trough concentration at steady state; C_avg,ss_: average steady-state plasma concentration; AUC: area under the concentration-time curve; T_max_: time to reach peak concentration; T_1/2_: half time; CL_t_: total clearance; CL_r_: renal clearance; V_d_: distribution volume; AI: accumulation index; f_m_: fraction of CMS converted to colistin in the body; SD: standard deviation. * *p* values were calculated by *t*-test comparing the parameters after the first and last doses. The bold numbers were considered as statistically significant (*p* < 0.05). ^a^ The median values of T_max_ after the first and the last doses were 1.37 h and 1.43 h for CMS and 5.00 h and 3.50 h for colistin, respectively. ^b^ The median values of T_1/2_ after the first and the last doses were 1.52 h and 1.72 h for CMS and 4.97 h and 6.10 h for colistin, respectively.

**Table 2 antibiotics-11-00798-t002:** Probability of target attainment (PTA) for colistin in terms of *f*AUC_0–24_/MIC after a CMS dose of 2.5 mg/kg CBA every 12 h for 7 consecutive days.

Bacteria Type	Effect	*f*AUC_0–24_/MIC Target ^a^	MIC of Colistin (μg/mL)						
			0.06	0.125	0.25	0.5	1	2	4
*Acinetobacter baumannii*	1 log kill ^b^	13.9	99.99	99.99	99.98	99.68	67.26	0.00	0.00
	2 log kill ^c^	17.6	99.99	99.99	99.96	97.69	24.01	0.00	0.00
*Klebsiella* *pneumoniae*	1 log kill ^b^	17.4	99.99	99.99	99.96	97.90	26.08	0.00	0.00
*Pseudomonas* *aeruginosa*	1 log kill ^b^	10.9	100.00	99.99	99.98	99.88	91.27	2.52	0.00
	2 log kill ^c^	13.7	99.99	99.99	99.98	99.68	69.49	0.00	0.00

^a^ Data from references [6,11]. ^b^ Colony count reduced to 10% of the baseline value after 24 h incubation. ^c^ Colony count dropped to 1% of the baseline value after 24 h incubation. The shaded PTA values were >90%.

**Table 3 antibiotics-11-00798-t003:** Comparison of PK/PD breakpoints and MICs of colistin against carbapenem-resistant Gram-negative organisms.

Pathogen ^a^	Effect	PK/PD Breakpoint	MIC of Colistin ^d^	
			MIC_50_	MIC_90_
CRAB	1 log kill ^b^	0.5	0.5	1
	2 log kill ^c^	0.5	0.5	1
CRKP	1 log kill ^b^	0.5	0.25	0.5
CRPA	1 log kill ^b^	1	1	1
	2 log kill ^c^	0.5	1	1

PK/PD breakpoints had the highest MIC with PTA ≥ 90% in Table 2. ^a^ CRAB: carbapenem-resistant *Acinetobacter baumannii*; CRKP: carbapenem-resistant *Klebsiella pneumoniae*; CRPA: carbapenem-resistant *Pseudomonas aeruginosa*. ^b^ Colony count reduced to 10% of the baseline value after 24 h incubation; ^c^ colony count dropped to 1% of the baseline value after 24 h incubation; ^d^ Adapted with permission from reference [12]. 2020, *Chin. J. Infect. Chemother*.

**Table 4 antibiotics-11-00798-t004:** List of individual adverse events related to the test drug CMS.

Subject ID	Clinical Adverse Event (Duration)	Abnormality in Laboratory Test
S01	Facial itchiness (2 days)	Normal
S02	Normal	Normal
S03	Tongue numbness (5 days and 20 h) Dizziness (4 h)	Normal
S04	Tongue numbness (5 days and 14 h)	Normal
S05	Tongue numbness (5 days and 8 h) Dizziness (2 days and 13 h) Headache (1 day and 19 h) Lip numbness (1 day and 10 h) Skin itchiness (1 day and 20 h)	Normal
S06	Tongue numbness (5 days and 20 h) Lip numbness (2 days and 15 h) Pharyngeal discomfort (5 days and 16 h) Facial numbness (4 days and 16 h)	Normal
S07	Facial itchiness (5 days and 4 h)	Normal
S08	Facial itchiness (4 days and 12 h) Tongue numbness (5 days and 2 h) Dizziness (2 days and 16 h)	Normal
S09	Facial itchiness (8 h)	Normal
S10	Tongue numbness (1 day and 20 h)	Normal
S11	Facial itchiness (2 days and 3 h) Tongue numbness (5 days and 8 h)	Normal
S12	Tongue numbness (6 days and 12 h) Lip numbness (5 days and 10 h) Sore throat (1 day and 23 h) Erythema (6 days and 13 h)	Decreased absolute value of neutrophils (2 days)

## Data Availability

All data generated or analyzed during the study appear in the submitted article.

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
