# Peer review of "Pharmacokinetics and Pharmacodynamics of Colistin Methanesulfonate in Healthy Chinese Subjects after Multi-Dose Regimen"

_antibiotics, 2022, doi:10.3390/antibiotics11060798_

Round 1

Reviewer 1 Report

This manuscript describes a study in which as used a multiple-dose regimen of colistin methanesulfonate (CMS) in human subjects. However, there are several aspects that should be clarified:

Abstract

Lines 24-25 and along with the manuscript the authors seem to refer that the Colistin methanesulfonate was administered every 12 hours for 7 consecutive days but this was confusing as the administration seems to be every 24 hours when we look for the figure 1 and table 1 for example. The concentration of CMS only presented maximum concentration levels after 24h, 48 h and so on, if the infusion was at every 12 hours, why the concentrations did not increase at the 12h, 36h, and so on?

  1. Introduction

Lines 77-82: the author's argument that in this study was used a new formula which was produced with a 4:1 ratio of CMS A: CMS B. However, when we compare it with the previous work of the same authors (reference 22 of this manuscript), it was evident that the previous formula was almost equal as in this article also a ratio of 4:1 has been reported. Which was the difference? 

If this new formula was compared with the identical experimental design manuscript previous work of the same authors (reference 18 of this manuscript), the main difference was only that the formula was slightly changed resulting in higher concentrations (and effects) however seems to remain insufficient to have effect against carbapenem-resistant A. baumannii. Probably an isobolographic analysis could be used to define the most benefic combinations of cholistins A and B.

  1. Results

Table 3: Regarding the evaluation of the MIC50 and MIC90 it seems that for CRPA the values were equal in both cases?!

2.4. Safety evaluation

The authors reported that no nephrotoxicity was observed but did not show any evidence for that conclusion.

  1. Discussion

A comparative analysis with other multiple-dose studies should be conducted in order to understand if in this study were presented better results compared with the other studies.

  1. Materials and methods

4.7. Safety evaluation

Almost nothing  was presented related to the biochemical evaluations in this manuscript (the values could be presented in a supplementary file).

Author Response

  1. Abstract

Lines 24-25 and along with the manuscript the authors seem to refer that the Colistin methanesulfonate was administered every 12 hours for 7 consecutive days but this was confusing as the administration seems to be every 24 hours when we look for the figure 1 and table 1 for example. The concentration of CMS only presented maximum concentration levels after 24h, 48 h and so on, if the infusion was at every 12 hours, why the concentrations did not increase at the 12h, 36h, and so on?

 Author’s response: Many thanks to the reviewers for the comments. The dose regimen was 2.5 mg/kg CBA for 7 consecutive days with 12 hours between each administration except for one dose in day 1 (first dose) and day 7 (last dose) (4.1. Study design and dosing regimen). At present, Figure 1 does give the impression that the dosing interval is 24 hours, so we have modified it to take into account the actual dosing and the measured concentrations.

  1. Introduction

Lines 77-82: the author's argument that in this study was used a new formula which was produced with a 4:1 ratio of CMS A: CMS B. However, when we compare it with the previous work of the same authors (reference 22 of this manuscript), it was evident that the previous formula was almost equal as in this article also a ratio of 4:1 has been reported. Which was the difference? 

 Author’s response: Reference 22 (now ref#10) is also a new ratio formula with CMSA: CMSB = 4:1, which is consistent with the present study. Reference 22 was a single-dose study that evaluated the effect of the difference in CMSA: CMSB component on single-dose PK parameters, whereas the present study was performed for PK and PK/PD analysis after a multiple dose. The description of the preliminary studies is added in the introduction (Line 78-82).

If this new formula was compared with the identical experimental design manuscript previous work of the same authors (reference 18 of this manuscript), the main difference was only that the formula was slightly changed resulting in higher concentrations (and effects) however seems to remain insufficient to have effect against carbapenem-resistant A. baumannii. Probably an isobolographic analysis could be used to define the most benefic combinations of cholistins A and B.

Author’s response: Many thanks to the reviewers for the comments, in future studies we will pay attention to some methods, such as isobolographic analysis to obtain the most beneficial combinations of colistin A and B.

  1. Table 3: Regarding the evaluation of the MIC50 and MIC90 it seems that for CRPA the values were equal in both cases?!

Author’s response: According to Ref #12 (Guo Y et al.), the MIC range of colistin against CRPA was 0.125-2 μg/mL, and the MIC values of colistin against 50% and 90% of bacteria were both 1 μg/mL.

  1. Safety evaluation

The authors reported that no nephrotoxicity was observed but did not show any evidence for that conclusion.

Author’s response: Supplementary Tables 1 for creatinine and early kidney damage indicators were added to the safety evaluation to describe the results.

  1. Discussion

A comparative analysis with other multiple-dose studies should be conducted in order to understand if in this study were presented better results compared with the other studies.

Author’s response: Comparisons with exposure in critically ill patients and healthy Japanese subjects after multiple dose administration are supplemented in the first and fourth paragraphs of the discussion, respectively.

  1. Materials and methods

4.7. Safety evaluation

Almost nothing was presented related to the biochemical evaluations in this manuscript (the values could be presented in a supplementary file).

Author’s response: These results are presented in supplementary Table S1.

Reviewer 2 Report

I read, Pharmacokinetics and Pharmacodynamics of Colistin Methanesulfonate in Healthy Chinese Subjects after Multi-dose Regimen, with interest. In this manuscript, the authors aim to investigate the PK characteristics of CMS and colistin in plasma and urine following multiple intravenous administrations of the new CMS formula at the recommended regimen in healthy Chinese subjects.

I have some questions and suggestions.

  1. Can you explain why this study is new or telling new things? Is it a novel study?
  2. Why was PTA > 90% selected as the target for dose selection in your study?
  3. Line 231-232: No obvious nephrotoxicity occurred during the study. Why do you not use criteria (RIFLE, AKIN) to assess nephrotoxicity?
  4. Demographic of the subjects: Please add baseline Scr of patients in this study.
  5. Results. I would read more information about serum β2-microglobulin, serum Cysteine dehydrogenase inhibitor C (Cystatin C), urine neutrophils gelatinase-associated lipocalin (NGAL), and urine β2-microglobulin of patients in this study. They could be responsible for a worsening of renal function.
  6. Line 191-205 in the Discussion section; ………….colistin combination treatment for invasive infections caused by CRO………. Please add more references and discussion for the colistin combination treatment of CRAB infection.

         please see,

      1. https://doi.org/10.3390/pharmaceutics13020162
  1. Could the absence of nephrotoxicity in this study be due to the short seven-day duration of colistin administration? Please discuss this issue.
  2. Discussion is rather weak. The data from other studies is relatively small, please include more PK/PD from other studies as well for comparison with this study.
  3. Line 228: Discussion section; … Nephrotoxicity is a major safety concern in the clinical application of CMS. Moreover, the administration of loading doses has increased the risk of colistin nephrotoxicity in some studies. Please insert and add more nephrotoxicity studies about loading dose colistin related with nephrotoxicity.

        please see (add),

      1. https://doi.org/10.1038/s41598-020-78098-4

       10.  Please write your conclusion more clearly.

Author Response

  1. Can you explain why this study is new or telling new things? Is it a novel study?

Author’s response: This study is the first to report the PK parameters of CMS and colistin in healthy Chinese subjects after application of CMS injection 2.5 mg/kg CBA every 12 hours (CMSA: CMSB =4:1) and to evaluate the effectiveness of the recommended dose of CMS injection (2.5 mg/kg CBA q12h) on CRO infection in the Chinese population based on pharmacokinetic/pharmacodynamic analysis.

2.Why was PTA > 90% selected as the target for dose selection in your study?

Author’s response: Based on the study by Frei et al. (PMID: 18252694), “PK-PD susceptibility breakpoints were defined as MICs with > or = 90% probability of target achievement.” Therefore, we also chose PTA ≥ 90% for dose selection in this study.

  1. Line 231-232: No obvious nephrotoxicity occurred during the study. Why do you not use criteria (RIFLE, AKIN) to assess nephrotoxicity?

Author’s response: The RIFLE and AKIN criteria are mainly applied to the diagnosis of acute kidney injury in patients. In this study, healthy subjects were included, and nephrotoxicity was monitored by early renal damage indicators and serum creatinine.

  1. Demographic of the subjects: Please add baseline Scr of patients in this study.

Author’s response: Scr was added in the demographic of the subjects.

  1. Results. I would read more information about serum β2-microglobulin, serum Cysteine dehydrogenase inhibitor C (Cystatin C), urine neutrophils gelatinase-associated lipocalin (NGAL), and urine β2-microglobulin of patients in this study. They could be responsible for a worsening of renal function.

Author’s response:  These results are presented in a supplementary Table S1.

  1. Line 191-205 in the Discussion section; ……colistin combination treatment for invasive infections caused by CRO………. Please add more references and discussion for the colistin combination treatment of CRAB infection.

         please see, https://doi.org/10.3390/pharmaceutics13020162

Author’s response:  In the Discussion section, the references of the pharmaceutics13020162 on colistin combination therapy for CRAB infection has been added (Ref# 20).

  1. Could the absence of nephrotoxicity in this study be due to the short seven-day duration of colistin administration? Please discuss this issue.

Author’s response: According to the study of Miano et al (doi:10.2215/CJN.06980717), colistin is associated with substantial excess AKI that is apparent within the first 72 hours of treatment. Therefore, the administration of CMS for 7 days is sufficient for the development of colistin-related nephrotoxicity. This issue is discussed in the Line 253-254.

  1. Discussion is rather weak. The data from other studies is relatively small, please include more PK/PD from other studies as well for comparison with this study.

Author’s response: A comparison of PK/PD on CRKP and CRPA infections from other studies has been added in the third paragraph of the discussion (Line 192-205).

  1. Line 228: Discussion section; … Nephrotoxicity is a major safety concern in the clinical application of CMS. Moreover, the administration of loading doses has increased the risk of colistin nephrotoxicity in some studies. Please insert and add more nephrotoxicity studies about loading dose colistin related with nephrotoxicity.

        please see (add), https://doi.org/10.1038/s41598-020-78098-4

Author’s response:  This recommended literature focuses on the efficacy of loading doses of colistin with carbapenems. We have added the following to the discussion of nephrotoxicity at loading doses in other studies: In addition, colistin loading doses can rapidly reach target concentrations, but Katip et al found that loading doses had a higher risk of nephrotoxicity than non-loading dose patients (adjusted HR, 1.57, 95% CI 1.14-2.17, P = 0.006). However, in a meta-analysis of eight clinical studies including 1115 subjects, no correlation between loading dose and increased nephrotoxicity occurred (RR = 1.31, 95% CI = 0.90-1.91).

  1. Please write your conclusion more clearly.

Author’s response:  The conclusion is revised as follows: In conclusion, the recommended multiple intravenous doses of CMS at 2.5 mg/kg CBA once every 12 hours for 7 consecutive days is safe in healthy Chinese subjects. After multiple administration, slight accumulation of the active form colistin was observed with the Cavg,ss at 1.27 ± 0.27 μg/mL. The PK/PD breakpoint of 0.5 µg/mL to kill 1 log CRKP or CRPA is greater than or equal to its MIC90, however, the PK/PD break-point of colistin to CRAB was ≤ MIC50 (1 μg/mL), suggesting the recommended dose of CMS may not be sufficient in treating some patients with high colistin MIC.

Reviewer 3 Report

The authors present the results from a single-center, open-label, clinical study regarding the administration of Colistin methanesulfonate in 12 healthy Chinese and the determination of CMS's PK profile along with its PD properties through Monte-Carlo simulations. The authors should be congratulated because the manuscript is well-organized and presented. The introduction is sufficient, materials and methods are well-described and the results well-presented and discussed (including the limitations of the study). 
A minor comment is that the authors could discuss a little bit more their findings with the EMA-FDA recommendations and data from other studies (ref 8). There are also some minor typos and grammar errors that should be addressed in the revision process. 

Author Response

  1. A minor comment is that the authors could discuss a little bit more their findings with the EMA-FDA recommendations and data from other studies (ref 8).

Author’s response:  The revised discussion of the EMA-FDA recommendations (Ref. 8) is as follows: Nation et al. found that the plasma Cavg,ss of colistin achieved in 51 critically ill patients with normal renal function (CLCr≥80 mL/min) was 1.29 (0.45-5.28) based on the FDA recommended dose (2.5-5 mg CBA/kg) (ref 8), and the median concentration in 214 patients with different renal functions was 2.35 (0.24-9.92) mg/L. The observed Cavg,ss in healthy subjects is quite similar to those are administrated according to the CMS algorithm in critical ill patients, suggesting that the current dosage can be applicable to the critical ill patients in China.

  1. There are also some minor typos and grammar errors that should be addressed in the revision process. 

Author’s response:  The typos and grammar errors were revised.

Round 2

Reviewer 1 Report

In general, the manuscript was improved and now should be accepted for publication.

Author Response

Thank you for the comments by the reviewer.